# Protocol for a pragmatic cluster randomised controlled trial assessing the clinical effectiveness and cost-effectiveness of Electronic RIsk-assessment for CAncer for patients in general practice (ERICA)

Willie Hamilton [1], Luke Mounce [2], Gary A Abel [3], Sarah Gerard Dean [4], John L Campbell,[5] Fiona C Warren [6], Anne Spencer,[7] Antonieta Medina-Lara,[8] Martin Pitt,[9] Elizabeth Shephard,[10] Marijke Shakespeare,[11] Emily Fletcher [12], Adrian Mercer,[11] Raff Calitri [11]

For numbered affiliations see end of article.

**Correspondence to**
Prof Willie Hamilton;
W.Hamilton@exeter.ac.uk

## ABSTRACT

**Introduction** The UK has worse cancer outcomes than most comparable countries, with a large contribution attributed to diagnostic delay. Electronic risk assessment tools (eRATs) have been developed to identify primary care patients with a ≥2% risk of cancer using features recorded in the electronic record.

**Methods and analysis** This is a pragmatic cluster randomised controlled trial in English primary care. Individual general practices will be randomised in a 1:1 ratio to intervention (provision of eRATs for six common cancer sites) or to usual care. The primary outcome is cancer stage at diagnosis, dichotomised to stage 1 or 2 (early) or stage 3 or 4 (advanced) for these six cancers, assessed from National Cancer Registry data. Secondary outcomes include stage at diagnosis for a further six cancers without eRATs, use of urgent referral cancer pathways, total practice cancer diagnoses, routes to cancer diagnosis and 30-day and 1-year cancer survival. Economic and process evaluations will be performed along with service delivery modelling. The primary analysis explores the proportion of patients with early-stage cancer at diagnosis. The sample size calculation used an OR of 0.8 for a cancer being diagnosed at an advanced stage in the intervention arm compared with the control arm, equating to an absolute reduction of 4.8% as an incidence-weighted figure across the six cancers. This requires 530 practices overall, with the intervention active from April 2022 for 2 years.

**Ethics and dissemination** The trial has approval from London City and East Research Ethics Committee, reference number 19/LO/0615; protocol version 5.0, 9 May 2022. It is sponsored by the University of Exeter. Dissemination will be by journal publication, conferences, use of appropriate social media and direct sharing with cancer policymakers.

**Trial registration number** ISRCTN22560297.

## STRENGTHS AND LIMITATIONS OF THIS STUDY

⇒ Improvements in primary care are seen as a key for improving early cancer diagnosis in the UK, and this trial is targeting that part of the diagnostic pathway.

⇒ This is a large, definitive trial, powered to identify a clinically important difference in cancer stage at diagnosis.

⇒ The trial is designed to minimise impact on participating practices, with outcome data being obtained from routinely collected National Health Service data.

⇒ One limitation is that the UK's national imperative to improve cancer diagnosis after the COVID pandemic may mean that use of other interventions (or eRATs themselves) is encouraged by policymakers, reducing the validity and reliability of the trial.

## INTRODUCTION

An estimated 10 000 UK cancer deaths each year would not occur if the UK matched the outcomes of other European countries.[1] Much of the difference is attributed to diagnostic delay.[2] The National Health Service (NHS) long-term plan, published in January 2019, specifically targets an increase in the percentage of patients with cancer whose cancer is stage 1 or 2 (thus potentially curable) at diagnosis to rise from the current 54% to 75% by 2028.[3] Diagnosis of cancer may occur by several routes, but the main ones are population screening, and diagnosis after symptoms have occurred. Although screening for cancer is effective for colorectal, breast, lung and cervical cancers,[4–6] less than 10% of total new UK cancers are identified by this route.

Most of the remainder are diagnosed after presenting with symptoms, usually to primary care. Of patients with cancer, just under 20% present with an emergency complication of their cancer; however, many of these patients have previously reported symptoms attributable to their cancer to primary care, but this presentation did not lead to a diagnosis of cancer.[7]

Within general practice, many studies have aimed at identifying the symptoms of possible cancer and quantifying their predictive value.[8] One main output has been risk assessment tools (generally abbreviated to RATs); these give precise estimates of the chance of an underlying cancer as a percentage figure. RATs provide precise estimates for single symptoms (eg, the risk of cancer of the lung for a person aged 40 years or more with haemoptysis is 2.4%), as pairs of symptoms (haemoptysis accompanied by loss of weight is 9.2%) or as repeated symptoms (a re-attendance with haemoptysis is 17%).[9] RATs are published for the 18 most common adult cancers, accounting for nearly 90% of the total cancer burden. These publications have been highly influential: in particular, they strongly contributed to the National Institute for Health and Care Excellence (NICE) guideline, 'Suspected cancer: recognition and referral (NG12)', which guides symptomatic diagnosis of cancer in the UK.[10]

The initial RATs, of paper, mouse mat, calendar, or web-based forms, increased cancer diagnostic activity,[11] although their impact on hard outcomes, such as stage at diagnosis or cancer survival, were unknown. Electronic RATs (eRATs) for seven major cancers (lung, colorectal, pancreas, oesophago-gastric, bladder, kidney and ovary) have been developed for the two largest UK primary care electronic healthcare record systems, SystmOne and EMIS, used in around 80% of English practices. The software performs daily calculations of individual cancer risk in patients aged 40 and over, using coded symptoms and laboratory results in the patient's record over the past year, and prompts the general practitioner (GP) when the risk of one or more of these cancers is equal to or above 2%. Some form of electronic clinical decision support for cancer diagnosis has been downloaded by practices and used by at least one practice member in approximately 12% of English practices.[12] Two systematic reviews recently concluded that more research evidence was needed for impact on time to diagnosis and treatment, stage at diagnosis and health outcomes, as well as research to understand how tools are used in GP consultations.[13] A feasibility trial of the oesophago-gastric eRAT published after these systematic reviews reported installation and regulatory problems that severely restricted use,[14] and a vignette study of the colorectal RAT suggested it changed the GP's inclination to refer in 26% of uses.[15]

One crucial aspect of eRAT research relates to cost-effectiveness: annual NHS spending on cancer diagnosis is approximately £1bn.[16] Observational data showed that increased use of the urgent cancer referral system improved survival,[17] but there are insufficient data to inform a cost-effectiveness analysis of the subject.[13]

## Objectives

The overarching aim of the trial is to assess the clinical and cost-effectiveness of using eRATs for six cancer sites—colorectal, lung, bladder, kidney, oesophago-gastric and ovarian cancers—compared with usual care for patients in general practice. Our hypothesis is that provision of eRATs will expedite the diagnosis of symptomatic cancer resulting in better cancer outcomes.

The primary objective is to compare the effects of using eRATs (vs usual care) on the percentage of patients with a newly diagnosed cancer at one of the six sites whose cancer is staged as being stage 1 or 2 (vs stage 3 or 4).

A secondary objective is to investigate differences in the stage at diagnosis of a further six cancers without eRATs (combined): breast, melanoma, prostate, non-Hodgkin's lymphoma, larynx and uterus. This is to investigate the possibility of an effect whereby eRATs are associated with increased diagnostic activity beyond the eRAT cancers. We will also investigate differences in the number of patients diagnosed with the six eRAT cancers combined, and the total number of cancers (excluding non-melanoma skin cancer) diagnosed, use of the 2-week wait referral system (the main pathway for urgent investigation of possible cancer in England) or equivalent for the six eRAT cancers combined, and across all cancers; the routes to diagnosis for each of the six eRAT cancers,[18] and for the six comparator non-eRAT cancers; the proportion of patients on a 2-week wait pathway receiving a diagnosis of cancer; whether a patient on a 2-week wait pathway has a diagnosis of cancer established (or refuted) within 28 days; 30-day and 1-year survival for those with cancer; the rate of cancer investigations—namely colonoscopies, sigmoidoscopies, upper gastro-intestinal endoscopies, chest X-ray examinations, abdominal ultrasound scans and abdominal CT scans. We will also conduct parallel cost-effectiveness analyses, service delivery modelling and a process evaluation.

## METHODS AND ANALYSIS
## Design and setting

The study is a pragmatic cluster randomised-controlled trial in England, in primary care medical practices using one of the two (SystmOne or EMIS) electronic record keeping systems. The clusters are practices, a term which includes single practices, and small groups of practices agglomerated administratively to single entities. These will be randomised 1:1 to receive either the intervention (access to the suite of eRATs) or usual care. Online supplemental appendix A shows pathways to a cancer diagnosis in the UK and illustrates how the intervention is expected to have an effect. It is unrealistic to offer eRATs to individual GPs, as there would be considerable contamination within any practice. Nevertheless, for a practice to be eligible to take part, we ask at least 50% of GPs in that practice to agree to use the eRATs. Although the intervention is at the practice level, some process and resource

use measures and all main trial primary and secondary outcomes relate to individual patients.

## Intervention

### The eRATS

The eRATs have been developed by a specialist IT team, Informatica systems Ltd, in partnership with the cancer charity, Macmillan. The risk estimates in the eRATs are from the original research papers for each cancer site.[9 19–24] Practices will access the software via a new cloud-based system called Skyline, specifically designed to facilitate efficient integration into GP clinical systems. CA marking of the Skyline version of eRATs was obtained in September 2021.

The eRATs have multiple functions. The first is the *'prompt'*. This collates relevant coded symptoms and blood tests in the patient's medical record from the previous 12 months, which are then assessed for the possibility of cancer, generating a risk score equivalent to the positive predictive value of the cancer features for each cancer. A prompt (pop-up), displaying the risk score(s), appears on screen when a registered user opens a patient's medical records and indicates that patient has a risk of 2% or higher for at least one of the studied cancers.

A second function is the *'symptom checker'*, allowing the clinician to add additional patient's symptoms to the eRAT checklist on screen; this process automatically recalculates the risk of any of the six cancers. On reviewing the risk score from the prompt and/or symptom checker, the clinician then decides the best course of management, which may be: (i) clinical review in primary care; (ii) ordering of test/investigations or (iii) referral to secondary care. Embedded within all eRATS are links to authoritative guidance regarding the early diagnosis of cancer, NICE NG12,[10] Macmillan's abbreviated NICE guidance,[25] and Cancer Research UK guidance.[26] These sources of information are added to assist management of the patient, but the decision whether or not to investigate is for the clinician and patient. Some EMIS practices also have access to the QCancer risk tool,[27] although embedded in a dormant state within the practice IT and record system, and requiring manual activation before operation. All practices will be asked not to use it during the trial.

### Justification of cancer sites

RATs are available for 18 adult cancers, each varying in their incidence, ease of diagnosis, amenability to treatment and proportion presenting as an emergency.

We elected to study cancer sites (a) which were in the top 15 cancers by incidence; (b) for which curative treatment is reasonably possible in symptomatic patients[28] and (c) with a significant percentage of patients presenting as an emergency.[29] Using these criteria, six cancer sites were selected, amounting to approximately half of all incident cancers. The selected six were: lung, colorectal, oesophago-gastric, ovary, kidney and bladder. The remaining nine cancers were considered as comparators to examine any practice level effect of increased cancer diagnostic activity. Three of these nine cancers, brain, pancreas and leukaemia, were removed for clinical and practical reasons: no eRAT is available for brain or leukaemia; in both brain and pancreas, symptomatic diagnosis is considered to have a very small likelihood of improving survival,[28] and in leukaemia, a full blood count (easily available in primary care) will usually establish the diagnosis, making an eRAT unlikely to expedite the diagnosis.[30]

### Training practices in using eRATS

Training in the use of the eRATs uses short, prerecorded videos available online coordinated by a practice 'research champion'. These videos show GPs how to use the prompt and symptom checker functions.

### Duration of intervention

Practice recruitment started in August 2019 and is expected to finish at the end of March 2022, including installation of the eRATs software. The trial was paused for 6 months in March 2020 owing to COVID-19. The formal start of the intervention window will be 1 April 2022 (although some practices might have delayed installation) and will close for all intervention practices on 31 March 2024.

### Usual care

Patients presenting to the control practices will experience the GP's usual diagnostic approach. GPs in control practices will have no specific on-screen prompt, although they may have access to hard copy (eg, paper or mouse mat) versions of the RATs, or to other cancer tools such as those supporting structured follow-up of symptomatic patients not selected for initial investigation. For EMIS practices with QCancer dormant in the system, control practices are expected to leave it dormant. We will document control practice use of RATs, other decision support tools, and access to and use of eRATs via interim and exit questionnaires completed within the first 12 months of a practice commencing the intervention and at the end of the trial. In line with intervention practices, trial time will formally begin for control practices on 1 April 2022 and end on 31 March 2024.

### Data collection window

Outcome data for all practices will be obtained for the 2-year period from 1 June 2022 to 29 May 2024. This data collection window is lagged behind the trial time window (1 April 2022 to 31 March 2024) in order to: (a) provide some time for practices to become accustomed to how the intervention functions prior to data collection and (b) to have a 2-month window following the end of the intervention window in order to allow cancers to be diagnosed in patients seen towards the end of that window.

### Sample size

There are around 130 000 new diagnoses of the six included cancers in the UK annually.[31] As each of our

six cancer sites has different proportions diagnosed at an early stage, the sample size calculation is based on a relative improvement in staging, using an OR of 0.8 for a cancer being diagnosed at stage 3/4 in the intervention arm compared with the control arm. This difference is quite large and equates to an absolute reduction of 4.8% in the intervention arm as an incidence-weighted figure across the six cancers. A much smaller improvement would still be clinically valuable but would necessitate an impossibly large trial.

For the inflation factor we have used an intracluster correlation coefficient based on our previous work, of 0.05.[32] An average cluster size of 23 patients with a diagnosed cancer with recorded stage during 2-year follow-up is expected, with a coefficient of variation for cluster size of 0.7, giving a design effect of 2.66. For an individually randomised trial with 90% power and an α threshold of 0.05, the sample size would be 2049 patients per arm. Adding in the design effect, this becomes 5497 patients, requiring 239 practices per arm, and 478 practices in total. Owing to changes in practice structure (such as practice mergers, closures or divisions), we anticipate the loss of up to 10% of recruited practices over the course of the trial; to account for this we will recruit a target of 530 practices overall, expecting 12 190 patients to be diagnosed with cancer in total.

### Practice recruitment

A total of 530 primary care practices across England will be recruited, supported by the National Institute for Health Research Clinical Research Network and strategic media releases to raise awareness of the trial. Practices that are proposing a split or a merger are not eligible for the trial, as the practices before or after the change night have been allocated to different arms in the trial. A method for identifying and managing unanticipated splits or mergers during the active phase of the trial is shown in online supplemental appendix B.

Patients are not being recruited into this trial—patient consent is not being sought for the use of the eRATs during the consultation. This is because eRATs are essentially an extension and enhancement of existing diagnostic tools already available to the GP to support their clinical decision-making. Other randomised controlled trials of interventions in primary care have taken this approach,[33] including the feasibility trial of the oesophago-gastric eRAT.[14 34 35] To promote patient awareness of the practice's participation in the Electronic RIsk-assessment for CAncer (ERICA) trial, including requesting practices to add it to their websites and any social media feed. A selection of patients will be recruited to the nested process evaluation and health economics studies (see below and online supplemental appendices C and D).

### Randomisation

Practices will be randomised using a 1:1 ratio into one of two trial arms: usual diagnostic care (control) and usual diagnostic practice plus access to the suite of eRATs, as the intervention. Randomisation will be computer-generated and web-based, conducted by an independent member of staff at the Exeter Clinical Trials Unit (ExeCTU), overseen by the CTU statistician (not the trial statistician). To promote balance between the trial arms in practices' use of the 2-week wait system, and therefore propensity to refer to secondary care, we will minimise randomisation by age–sex standardised 2-week wait referral ratio (the best available proxy) in national tertiles. We will use simple randomisation to allocate the first 50 practices (~10% of the total target), and then apply minimisation by 2-week wait referral ratio tertile, taking into account the previous allocations to inform the minimisation algorithm. All allocations using the minimisation algorithm will retain a stochastic element, aimed at promoting allocation concealment.

The data analysis will be carried out by the trial statistician and health economist, blinded to treatment allocation, and all primary outcome data are objective assessments of clinical outcome. Staging (the primary outcome) will be performed by pathologists unaware of trial participation or allocation. However, given the nature of the intervention, it is not possible to blind GPs or the GP practice to treatment allocation.

### Outcome measures

#### Primary outcome

Outcome measures will be captured at patient level, using data routinely collected by the National Cancer Registration and Analysis Service (NCRAS). The primary outcome is whether a patient is diagnosed at stage 1 or 2 (early) or stage 3 or 4 (advanced). This division of staging is commonly used and is a targeted metric in the 2019 NHS Long Term Plan—for stage 1 and 2 cancers (for all staged cancers other than non-melanoma skin cancer) at diagnosis to comprise 75% of the total by 2028. The current UK overall incidence-weighted percentage of early stage at diagnosis was 55% in 2018, though for the six eRAT cancers, it is 35%.[36]

#### Secondary outcomes

A range of secondary outcomes will be examined:
► The binary stage at diagnosis of a further six cancers without eRATs will be identified from NCRAS, and compared between intervention and control practices. This is to investigate the possibility of a 'spillover' effect whereby eRATs are associated with increased diagnostic activity beyond the eRAT cancers.
► The practice's number of patients diagnosed with the six eRAT cancers combined, and the total number of cancer cases, from NCRAS.
► The number of patients investigated or referred under the 2-week wait system for the six eRAT cancers combined, and in total, from Cancer Waiting Times data.
► Route to diagnosis from the Routes to Diagnosis Dataset,[18] which uses Hospital Episode Statistics data. This will be categorised into four possible

routes: emergency attendance, 2-week wait referral, GP referral and 'other'. We will collect this information for each of the six eRAT cancers, and for the six comparator non-eRAT cancers.

► The 2-week wait performance measures, from Cancer Waiting Times data, for the six eRAT cancers combined, and for all cancer referrals:
  – Whether a patient on a 2-week wait pathway received a diagnosis of cancer. When aggregated, for example at the practice level, and expressed as the proportion of patients who received a cancer diagnosis, this is known as the conversion rate.
  – The duration between 2-week wait referral and diagnosis of cancer in days.
  – Whether patients referred on a 2-week wait referral and who received a cancer diagnosis were diagnosed within 28 days, the Faster Diagnosis Standard (introduced in 2022).
  – Detection rate—the proportion of a practice's cancers which are identified via the 2-week wait pathway.

► Survival measures (from date of diagnosis): 30-day; 1-year (identified from NCRAS). 5-year survival will also be reported, but the main trial will report at 30 days and 1 year, with 5-year data being a subsidiary report. These outcomes will use all-cause mortality data from the Office for National Statistics.

► Adverse events (using data from the Diagnostic Imaging Dataset): these are expected to be few, and largely related to complications from hospital investigation, such as colonoscopy. There is no mechanism for adverse events to be collected using routine data. We will, however, estimate any change in the expected number of adverse events from imaging investigations (colonoscopies, sigmoidoscopies, upper gastrointestinal endoscopies, chest X-ray examinations, abdominal ultrasound scans and abdominal CT scans) through investigating any change in the rate of these investigations in intervention practices relative to control practices (see data analysis section). Potential adverse psychological consequences of being labelled with 'possible cancer' will be further explored in the process evaluation.

### Data collection
All primary and secondary outcome measures are available from NCRAS, Diagnostic Imaging Dataset and publicly available practice level data, including Cancer Waiting Times data. We will be using depersonalised (pseudo-anonymised) data. The Public Health England Office for Data Release guidelines indicated that no legal gateway (eg, section 251 approval) will be necessary to obtain these data.

### Data analysis
All analyses will follow Consolidated Standards of Reporting Trials (CONSORT) guidelines for cluster-randomised and pragmatic trials. The primary analysis,

exploring the proportion of patients with early-stage cancer at diagnosis, will use mixed-effects logistic regression with a random intercept for practice to accommodate the hierarchical nature of the data (ie, random allocation by practice, with participants nested within a practice). This regression will include trial arm at practice level, and will adjust for patient-level covariates known to be associated with stage (age, sex, quintile of the income domain from the Index of Multiple Deprivation (IMD) and cancer site),[37] and the practice-level minimisation variable (national tertile of age–sex standardised 2-week wait referral ratio). We will further adjust the model at the practice level for list size, clinical IT system used, and Care Quality Commission overall rating, should these variables be associated with stage in preliminary analyses (even if not unbalanced with respect to trial allocation). Trial arm and covariates will all be entered as fixed effects. The degree of change in the percentage of patients diagnosed at a late stage in intervention practices will be investigated by exploring the marginal distributions of trial arm on the probabilities predicted by these models.

For the secondary outcome of the stage at diagnosis of six cancers without eRATs, we will repeat the above model including data on the six non-eRAT cancers as well as the six eRAT cancers. This model will use all the variables described above, plus an indicator variable for whether the cancer site has an eRAT, and an interaction term between this variable and trial arm. From this model, we will obtain odds ratios (with 95% CIs) for (i) the 'spillover' effect of having the intervention on cancer sites not included in the intervention, and (ii) for the relative effect of the intervention on stage for included cancer sites compared with those not included in the intervention.

Mixed-effects logistic regression models with a random intercept for practice will also be fitted for the other secondary binary outcomes; route to diagnosis, conversion rate and timeliness. These models will include trial arm as a practice-level effect, and will adjust at the patient level for age, sex and quintile of the IMD income domain, and at the practice-level for the minimisation variable (national tertile of age–sex standardised 2-week wait referral ratio). These analyses will also adjust at the patient level for cancer site (routes to diagnosis analyses) or for referral type (2-week wait analyses) as appropriate. The models will be further adjusted as in the main outcome variable analysis.

Time-to-event secondary outcomes (length of waiting time, survival) will be analysed using mixed-effects parametric survival models with a random intercept for practice, and all other variables added as fixed effects. These models will include trial arm as a practice-level effect, and will adjust for the same patient-level factors as described above (waiting times adjusted for referral pathway rather than cancer site as above), and the practice-level minimisation variable (national tertile of age–sex standardised 2-week wait referral ratio). The models will also use the same adjustment as the primary outcome measure. An

appropriate distribution to model the baseline hazard will be used, as determined by a comparison of the Akaike information criteria under different distributions.[38]

For rate outcomes (number of 2-week wait referrals, cancers and imaging investigations), we will analyse the rates per 100 000 registered patients per year by age–sex strata using mixed-effects Poisson regression models including a random intercept for practice. These models will include trial arm as a predictor and will adjust for the age and sex of the strata, and at the practice level for the minimisation variable (2-week wait referral ratio) and deprivation (quintile of IMD overall score). The models will be further adjusted at the practice level for list size, clinical IT system used, Care Quality Commission overall rating, and for the age and sex case mix of practices should these covariates be found to be associated with the outcome (even if not unbalanced with respect to allocation). Case mix will be incorporated by including variables for counts of practice populations in different age–sex strata (5-year age groups by sex, excluding one age–sex stratum that can be determined once all others are known).

All the above analyses will combine data for the six eRAT cancers for each model. For outcomes related to 2-week wait referrals, data will be combined for all referral pathways relevant to the six eRAT cancers. To investigate whether the eRATs produce a spillover effect, whereby diagnostic activity is increased for other cancers, we will repeat all analyses using data for the six non-eRAT cancers combined for each model. Investigation of a spillover effect for 2-week wait referral outcomes will use data for all referral pathways combined.

Additional sensitivity analyses will be conducted for the primary outcome in order to explore moderation arising from practice-level characteristics, using interaction terms. Although the trial has not been powered to detect low-to-moderate subgroup differences, such as differences in a single cancer site, large interaction effects that differ with respect to the direction of effect across subgroups are of interest. The potential impact of missing staging data on the primary outcome will also be explored through use of multiple imputation methods making use of auxiliary variables, such as survival time, morphology and grade, to improve the missing at random assumption in line with previous work).[37]

### Data management
Cancer registry data (NCRAS) will be managed and prepared by the registry themselves and securely, electronically transferred to the study team. There will be no patient identifiable data within these datasets. Data from NCRAS will be stored on the secure data resource hub at the University of Exeter (which meets requirements for secure storage of sensitive data) and linked to existing practice data held within ExeCTU's REDCap database. The data will be stored and retained in accordance with registry policies.

The nested studies rely on identifying patients from in-practice usage reports. These reports contain depersonalised (pseudo-anonymised) data. The practice will send a copy to the trial team with the original practice ID number removed. The local in-practice reports will be securely and electronically transferred to a secure Exeter CTU computer.

In the recruitment of patients (and NHS staff) for interviews, questionnaires or permission for access to medical notes, participant details will be passed securely between NHS services and the research team. All participants agreeing to interview, to complete a questionnaire and/or medical notes review, and all GPs agreeing to interview will be allocated a unique study ID, and the information linking their ID to their personal details will be kept securely at the University of Exeter. All other participant-related paper records will be anonymised and stored separately from the personal information. The electronic database for the trial will be stored on the secure servers of the University of Exeter with password-controlled access provided for the research team by ExeCTU. Single data entry with extensive in-built validity checks will be used to reduce the risk of transcription errors.

Audio recordings will be digitised, encrypted and stored on the university's secure server. Audio recordings will be retained until after anonymised transcripts have been finalised and analysed. At this stage they will be securely and permanently deleted. Access to personal data will be restricted to the research team. Names and participant details will not be passed to any third parties and no named individuals will be included in the outputs. All participants (patients, NHS staff) will be asked for their consent for the study team to retain interview transcripts for the purposes of future research by those involved directly in the study team or to be used for educational purposes.

Informatica Systems Ltd has developed a separate agreement ('Data processing deed') for intervention practices, which will be used between the GP practices and Informatica Systems Ltd. The deed was necessary because the development of Skyline has affected the processing arrangements for the eRATs software that is used. The ERICA research study will still use the organisation information document which outlines the research team's data processing requirements, to be signed between the practice and sponsor.

All study data will be kept for 10 years (unless data registry policy requires otherwise) under secure conditions on University of Exeter secure servers. Data will also be subject to standard secure storage and usage policies.

### Trial monitoring and management
#### Trial sponsor and funders
The University of Exeter is the trial sponsor. The trial funders are providing finance to run the trial. None of the funders or sponsor will be involved in the design or day-to-day conduct of the trial, analysis of data or interpretation of findings.

## Trial steering committee (with data monitoring committee responsibilities)

The responsibilities of the trial steering committee (TSC) will be to review the main study protocol and any amendments, monitor and supervise the trial towards its interim and overall objectives, review relevant information from other sources and help to resolve problems brought by the trial management group (TMG). The TSC will therefore provide overall independent supervision for ERICA on behalf of the funders and the sponsor. Meetings will be held at regular intervals determined by need and not less than twice a year. Routine business will be conducted by telephone, videoconference, and email. The TSC will also operate as a data monitoring committee with responsibility for the overall conduct of the trial. There will be a time lag between practices 'entering the trial' and data availability from cancer registries. The time lag will be such that data will be available only when practices have completed data collection. Therefore, interim analyses to assess whether the trial was effective, and to support a decision whether to stop the trial early, would be unnecessary as data collection (and practice participation) would have already ceased.

## Trial management group

A TMG has been established and includes those responsible for the day-to-day management of the trial and those supporting the delivery of the trial and associated stakeholders, including representatives of the local clinical research networks and Macmillan. The group will monitor all aspects of the conduct and progress of the trial, ensure that the protocol is adhered to and take appropriate action to safeguard participants and the quality of the trial itself. The group will meet regularly (monthly in the first instance, until recruitment has completed) in person and/or by phone or over the internet (via Microsoft Teams).

## Core study team

The core study team (chief investigator, trial manager (TM)) will meet weekly during the study. Day-to-day running of the trial will be the responsibility of the TM. The TM will have access to the ExeCTU suite of standard operating procedures (SOPs) and will ensure that the trial is run in compliance with all relevant SOPs (eg, assessment, processes and reporting, data management, study staff health and safety).

## Nested studies
### Health economics

We will estimate the cost and cost-effectiveness of the eRATs versus usual diagnostic practice using the primary perspective of the NHS and personal social services (ie, third-party payer). We will estimate the cost-effectiveness of the intervention based on the primary outcome and secondary survival outcomes (30 day and 1 year; 5-year survival will be a subsidiary report) for the six cancer sites with eRATs and report the results using the latest guidelines.[39] For colorectal, lung and ovarian cancers we will use decision analytic models to combine data from the within-trial analysis of ERICA intervention on costs and benefits, with longer estimates derived from the evidence synthesis of the costs and benefits of stage of diagnosis and disease progression to estimate the cost per quality-adjusted life-year over the longer term.[40] For fuller details see online supplemental appendix C.

### Service delivery modelling

This will investigate the key factors central to the (re) organisation of NHS diagnostic services for cancer referrals. We will use a range of methods, both quantitative and qualitative, to analyse service delivery alternatives. Specifically, we will aim to use modelling approaches to explore the likely implications of different scenarios across dimensions of performance, outcomes and costs. Fuller details are in online supplemental appendix D.

### Process evaluation

The process evaluation work aims to identify and investigate the contextual factors that impact on the effectiveness of the eRATs, with particular focus on intervention fidelity and GP engagement. The impact of the eRATs on the patients' experience of their GP consultation and their experiences of subsequent care will also be explored. Fuller details are in online supplemental appendix E.

### GP workload

This nested study aims to explore, in terms of consultation time, the impact of the use of eRATs by GPs on their workload, and patient 'flow' through consulting sessions. It will also explore workload in the week following the index consultation in which an eRAT was activated. Fuller details are in online supplemental appendix F.

## Patient and public involvement and engagement

Our patient and public involvement and engagement (PPIE) group, including cancer survivors, has been consulted widely during the development of this study. The PPIE group has reviewed and commented on the protocol and supported the development of all patient-facing materials, including information sheets and study lay summaries. One experienced PPIE representative sits on the TMG and another is on the TSC. A total of seven people have joined our PPIE group for this study and will contribute by reviewing study materials and documentation, commenting on and proof reading reports and contributing to dissemination activities. This group will be supported in their work by the South West Peninsula applied research collaboration PPIE team—for example, by attending workshops on critical appraisal skills. All PPIE representatives will be recompensed for their time given to the study.

## Ethics and dissemination

A trial publication policy will be developed which outlines the plan for dissemination and will be in accordance with the International Committee of Medical Journal Editors.

The results of the trial will be reported first to study collaborators and to the funder. The main report will be drafted by the TMG and circulated to all collaborators and the TSC for comment.

Access to the final trial datasets will be made publicly available unless contractual agreements between data providers limit such access.

### Ethical review

The trial has received favourable ethical review from London City and East Research Ethics committee, reference number 19/LO/0615, with eight amendments between then and 2022, relating to three main areas: the delays caused by the COVID-19 pandemic, with its recruitment moratorium; an alteration in the mechanism by which the eRATs software was delivered; and the inclusion of a nested study focusing on the impact of eRATs on GP workload. Current protocol version – V 6.0, eighth.[39]

**Author affiliations**
[1]Primary Care Diagnostics, University of Exeter, EXETER, UK
[2]Institute of Health Research, University of Exeter, Exeter, UK
[3]University of Exeter Medical School (Primary Care), University of Exeter, Exeter, Essex, UK
[4]PenCLAHRC University of Exeter Medical School, Exeter, UK
[5]Primary Care, University of Exeter, Exeter, UK
[6]Institute of Health Research, University of Exeter Medical School, Exeter, UK
[7]Health Economics, University of Exeter Medical School, Exeter, UK
[8]Health Economics Group, University of Exeter Medical School, Exeter, UK
[9]University of Exeter: Medical School, University of Exeter, Exeter, Essex, UK
[10]University of Exeter, Exeter, UK
[11]Primary Care, University of Exeter Medical School, Exeter, UK
[12]Primary Care Research Group, University of Exeter Medical School, Exeter, UK

**Acknowledgements** We thank the National Institute for Health Research (NIHR) Clinical Research Network for their support with recruitment, Macmillan for their contributions to the early electronic risk assessment tools work and ongoing support with practice recruitment and pilot testing. SGD's time is partially supported by the National Institute of Health Research Applied Research Collaboration (ARC) South-West Peninsula. Disclaimer: The views expressed are those of the authors and not necessarily those of the NHS, the NIHR or Department of Health.

**Contributors** WH conceived of the trial. Substantial contributions to the design of the methods and research processes were made by WH, JLC, LM, SGD, GAA, MP, AS, AM-L, FCW, EF, ES, MS, AM and RC. The protocol was written by RC, LM, SGD, GAA, AS, EF, and MP under the overall editorial control of WH. All authors critically reviewed the protocol and provided approval of the final version.

**Funding** This research is funded by a philanthropic donation by The Dennis and Mireille Gillings Foundation, Cancer Research UK (C8640/A23385), plus support from Macmillan in provision of staff time, and the University of Exeter. The trial is registered with ISRCTN: (trial no: ISRCTN22560297) and on the CRUK trial registry (CRUK database no: 16163).

**Competing interests** WH owns background IP for the eRATs but will not use this for personal finacial benefit from any NHS use. All other authors declare no competing interests.

**Patient and public involvement** Patients and/or the public were involved in the design, or conduct, or reporting, or dissemination plans of this research. Refer to the Methods section for further details.

**Patient consent for publication** Not applicable.

**Provenance and peer review** Not commissioned; externally peer reviewed.

**Data availability statement** Data sharing not applicable as no datasets generated and/or analysed for this study. Data sharing not applicable as no datasets generated and/or analysed for this Protocol paper.

**ORCID iDs**
Willie Hamilton http://orcid.org/0000-0003-1611-1373
Luke Mounce http://orcid.org/0000-0002-6089-0661
Gary A Abel http://orcid.org/0000-0003-2231-5161
Sarah Gerard Dean http://orcid.org/0000-0002-3682-5149
Fiona C Warren http://orcid.org/0000-0002-3833-0182
Emily Fletcher http://orcid.org/0000-0003-1319-3051
Raff Calitri http://orcid.org/0000-0003-0889-4670

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
