## [Reviewer comments · BMJ Open]

ARTICLE DETAILS

TITLE (PROVISIONAL)	Protocol for a pragmatic cluster randomised controlled trial assessing the clinical effectiveness and cost-effectiveness of electronic risk-assessment for cancer for patients in general practice (ERICA)
AUTHORS	Hamilton, Willie; Mounce, Luke; Abel, Gary; Dean, Sarah; Campbell, John; Warren, Fiona; Spencer, Anne; Medina-Lara, Antonieta; Pitt, Martin; Shephard, Elizabeth; Shakespeare, Marijke; Fletcher, Emily; Mercer, Adrian; Calitri, Raff

VERSION 1 – REVIEW

REVIEWER	Seema Khan Feinberg School of medicine of Northwestern, Surgery
REVIEW RETURNED	03-Oct-2022

GENERAL COMMENTS	The authors describe a protocol which is been funded and approved for initiation, with the purpose of shifting cancer diagnosis in England to an earlier stage. The primary goal is to increase the proportion of cancers diagnosed in stages one and two rather than three and four. The plan is to cluster-randomize primary care practices to an intervention of electronic risk assessment tools based on symptoms that are potentially related to malignancy and compare the fraction of early-stage cancers diagnosed in these practices compared to practices pursuing usual care. The research plan is well-designed, and well described. But the clarity would be greatly aided by a chart showing patient flow through primary care practices in the UK, in practices randomized to the intervention versus those who are not. This would be particularly helpful for readers who are not familiar with UK systems. The hypotheses are not clearly enunciated. Given the number of objectives, and the number of possible pathways for patients present with symptoms that may be related to cancer, it would be helpful to see clearly stated hypotheses that orient the reader to the specific endpoints to be analyzed, particularly given the number of secondary objectives, some of which are quite “soft”. The secondary outcomes are quite extensive, and not all are clearly defined. In particular, objective 5.1 is difficult to follow. “Whether a patient on a 2-week wait pathway received a diagnosis of cancer, expressed as– the proportion of patients who received a cancer diagnosis, also known as the conversion rate”. It appears that each patient (the numerator of one) will be used to generate a proportion. This does not make sense. For 5.2, please clarify whether the 28 day interval to cancer diagnosis begins at the time that the patient is seen by the GP and is referred into the two week track, or begins at
--

	the end of the two week track. Secondary Objective six relates to survival, with 30 day and one year survival being the main measures. The cancers being targeted are colorectal, lung, bladder, kidney, oesophago-gastric and ovarian cancers. What is the starting event? GP visit, initiation of cancer directed diagnostics, or actual diagnosis date? Death from cancer within 30 days seems unlikely for these primary sites, except for the most advanced cancers, and it is not clear why this is considered relevant to the objectives of the trial. Presumably, these early deaths are rare, and would be too few to see differences between intervention and usual care practices.
--	---

REVIEWER	Hayley Thomas University of Queensland Faculty of Medicine and Biomedical Sciences, Primary Care Clinical Unit
REVIEW RETURNED	29-Nov-2022

GENERAL COMMENTS	Thank you for the opportunity to review this interesting work. The proposed concept sounds novel to my knowledge, and potentially promising to improve cancer diagnosis. The protocol is comprehensive, well written and largely clear. I have minor suggestions to improve clarity in parts and some thoughts regarding potential issues with the intervention and additions to the methodology that the authors may like to consider. Please note that I am unable to provide expertise regarding the statistical robustness of the protocol. Regarding the intervention:  -If I have understood correctly, the eRAT prompt relies on data coded in the medical record. In my experience, GPs do not always code symptoms in practice management software (despite recording these in progress notes). This may lead to failure to generate a prompt, limiting the effectiveness of the intervention. I imagine the researchers are aware of this potential issue; are there measures in place to support/encourage GPs in the intervention group to consistently code patient symptoms? -For intervention practices, I wonder whether GPs may be falsely reassured if an eRAT prompt is not generated for a patient with cancer, counterintuitively leading to diagnostic delay? This study as it is planned should, however, help to answer that question. Regarding the study methodology:  -Page 7, "Sample size": I agree with the authors that the absolute reduction of 4.8% in stage 3/4 cancer diagnoses is quite large; and that it is unfortunate that detecting a smaller improvement is not feasible due to sample size required. -Page 10, point 7 "Adverse events": In addition to harms from investigations and psychological distress, it may be appropriate to consider the potential harms of cancer over-treatment in this analysis (see, for example: https://www.racgp.org.au/download/Documents/Guidelines/prostate-cancer-screening-infosheetpdf.pdf) -In addition to comparing the stage at cancer diagnoses for all 6 eRAT cancers combined, will subgroup analysis also be performed for each of the 6 cancers individually? It would be interesting to see if there was a difference in efficacy among the 6 eRATs. For example, it may be that for some cancers symptoms occur primarily in stage 3 or 4 disease, making the intervention less effective.
---

-As early detection of malignancy does not always confer mortality benefit, the secondary outcome of survival is perhaps the most important outcome for this study. For example, a recent trial that concluded that despite earlier detection of ovarian cancer with screening (using CA125 levels or transvaginal ultrasound), this did not translate to a mortality benefit (Menon U, Gentry-Maharaj A, Burnell M, et al. Ovarian cancer population screening and mortality after long-term follow-up in the UK Collaborative Trial of Ovarian Cancer Screening (UKCTOCS): a randomised controlled trial. Lancet. 2021;397:2182-93. 33991479).

-Page 22, Appendix A: There is a risk of underestimating the effectiveness of the intervention by classifying mergers of control/non-trial practices + intervention practices as intervention practices. I imagine that the authors have considered this and decided that the benefit of avoiding reduction in the study power justifies this risk?

-Page 23, Appendix B: Is there a rationale for the estimated 40% patient acceptance rate to respond to the survey? I wonder if this might be a little generous?

-Appendix D: Will GPs be remunerated for their time to participate in training? This may increase engagement?

-Page 33, Appendix E: Does a GP consulting session in England usually include 24 back-to-back consultations without any slots deliberately remaining unbooked (for administrative work, catch up time etc)?

-Page 34, Appendix E, Primary Outcome: The duration that a GP keeps a chart open doesn't always correlate with consultation length (some GPs keep charts open to write notes later etc).

-Page 35, Appendix E, Measuring duration of consultations: There may be differences in complexity for patients who trigger eRATs vs those who do not that are unrelated to the eRAT prompt (ie. eRAT prompt patients may be more unwell than others and therefore require more time). Rather than comparing time of consultations with eRAT trigger vs no eRAT trigger in intervention practices, it would be more accurate to compare time of consultations where an eRAT is triggered in intervention practices with time of consultations where an eRAT would have been triggered in control practices. If this is not possible, data analysis should adjust for patient factors known to affect consultation length as well as for factors such as consulting GP, time of day etc.

Points for clarification/minor edits:

-Page 5, first line: I don't quite understand the 'symptom checker'. Are you saying that the computer brings up the completed eRAT checklist that generated the prompt, and the clinician checks which symptoms it has identified and manually edit this if necessary?

-Page 5, "Justification of cancer sites": This is slightly confusing. The first line states that 18 RATs are available. The third line goes on to mention the 15 most common cancers. It isn't immediately clear (until I added up the numbers) whether the 'six cancers selected' (line 5) and 'nine remaining cancers' (line 6) are referring to the 18 RATs or the 15 most common cancers. Perhaps make a slight edit on line six to clarify – 'The remaining nine of the fifteen most common cancers were considered...'. I would also suggest listing the six cancer sites that were selected for eRATs in this section.

-Page 8, "Practice Recruitment": The sentence beginning 'To promote practice awareness...' requires grammatical correction.

-Page 14, "Nested Studies, Health Economics": 'For three cancer sites we will use decision analytic models...' Which 3 cancer sites

	are you referring to? -Page 21, "Appendix A". Missing word in first sentence of third paragraph – "Excluding practices who restructure..." -Page 22, "Appendix A": "Where changes in list size...from one month prior to the drop" – A change could be a drop or alternately an increase in size – please clarify whether this approach will be applied for any change (increase or decrease) or only for a drop in size? -Page 23, "Appendix B": Note incomplete sentence "With a conservative estimate of a cluster size of five patients responding to the questionnaire." -Page 26, "Appendix D": Unbalanced parentheses– '(i.e., ongoing use of the eRATs...' -Page 28, "Appendix D": Second paragraph, last line – "GPs" does not require an apostrophe -Page 29, "Appendix D": 4th paragraph, 6th line – suggest clarification - "(false positive eRAT result)" - currently not immediately obvious whether the cancer diagnostic test or the eRAT is the false positive. -Page 34, "Appendix E": Final line, remove "practices across the nested studies" (repetition) -Page 35, "Appendix E": First line, remove "only" (repetition) Thank you again for the opportunity to review this protocol. I would like to acknowledge my colleague, Dr David King, who also provided input into this review. I look forward to hearing the study outcomes.
--	--

REVIEWER	Walter Lehmacher University of Cologne, Medical Statistics
REVIEW RETURNED	20-Dec-2022

GENERAL COMMENTS	A well done design paper.
---------------------------

REVIEWER	Kuen-Cheh Yang National Taiwan University Hospital, Bei-Hu Branch, Taipei, Taiwan, Family Medicine
REVIEW RETURNED	29-Dec-2022

GENERAL COMMENTS	This review is mainly for the statistical part only. The vast majority of statistical methodology is already written well. I have some minor comments. #Survival analysis.  1. The author mentioned secondary survival outcomes of 30-day, 1-year and 5-year. However, the data collection window is 1/6/2022-29/5/2024. How would the survival status be collected? 2. The definition of survival: does the author mean the overall survival or specific cancer survival? #cost-effectiveness analysis (CEA)  1.For the decision analytical model, I would like to see more details regarding the decision tree. Then, it would be helpful to understand how the QALY 2.For probabilistic sensitivity analysis, I would like the author to provide more information: such as the distribution of costs, or what kinds of parameters would be samples. 3.Time horizon is also a key part in CEA. Can the author elaborate more on this?
---

VERSION 1 – AUTHOR RESPONSE

Reviewer: 1	
1.1 The authors describe a protocol which is been funded and approved for initiation, with the purpose of shifting cancer diagnosis in England to an earlier stage. The primary goal is to increase the proportion of cancers diagnosed in stages one and two rather than three and four. The plan is to cluster-randomize primary care practices to an intervention of electronic risk assessment tools based on symptoms that are potentially related to malignancy and compare the fraction of early-stage cancers diagnosed in these practices compared to practices pursuing usual care.	Thank you – no response required.
1.2 The research plan is well-designed, and well described. But the clarity would be greatly aided by a chart showing patient flow through primary care practices in the UK, in practices randomized to the intervention versus those who are not. This would be particularly helpful for readers who are not familiar with UK systems.	We have added a figure, in Appendix F, which outlines the usual pathway to a cancer diagnosis in the UK. This also helps to describe some of the metrics for the referee's next point. We reference this figure in the main text.
1.3 The hypotheses are not clearly enunciated. Given the number of objectives, and the number of possible pathways for patients present with symptoms that may be related to cancer, it would be helpful to see clearly stated hypotheses that orient the reader to the specific endpoints to be analyzed, particularly given the number of secondary objectives, some of which are quite "soft".	We have added a specific hypothesis statement Our hypothesis is that provision of eRATs will expedite the diagnosis of symptomatic cancer resulting in better cancer outcomes. (p.3) As to the various secondary outcomes, each is relevant to some aspect of the pathway. We have used the figure referred to above, which helps to explain each outcome.
1.4 The secondary outcomes are quite extensive, and not all are clearly defined. In particular, objective 5.1 is difficult to follow. "Whether a patient on a 2-week wait pathway received a diagnosis of cancer, expressed as– the proportion of patients who received a cancer diagnosis, also known as the conversion rate". It appears that each patient (the numerator of one) will be used to generate a proportion. This does not make sense. For 5.2, please clarify whether the 28 day interval to cancer diagnosis begins at the time that the patient is seen by the GP and is referred into the two week track, or begins at the end of the two week track.	We have amended the description of the secondary outcomes to improve readability (p.8-9). In particular, outcome 5.1 (conversion rate) will be reported at the practice-level using data aggregated from the patient-level. Hence the denominator is all patients from a given practice who received a 2-week wait referral, and the numerator is those who received a diagnosis, using patient-level data in the form of a binary indicator for whether they received a diagnosis. Outcome 5.2 accidentally combined two separate secondary outcomes, and we thank the reviewer for bringing this to our attention. 5.2 is now the time in days from referral to diagnosis; 5.3 is now whether diagnosis met the Faster Diagnostic Standard (28 days from referral to diagnosis); and 5.4 is the detection rate (was 5.3).
1.5 Secondary Objective six relates to survival, with 30 day and one year survival being the main measures. The cancers being targeted are colorectal, lung, bladder, kidney, oesophago-gastric and ovarian cancers. What is the starting event? GP visit, initiation	We now clarify that survival measures have their start point at date of diagnosis and relate to all-cause mortality. In an ongoing analysis of 288,297 English

of cancer directed diagnostics, or actual diagnosis date? Death from cancer within 30 days seems unlikely for these primary sites, except for the most advanced cancers, and it is not clear why this is considered relevant to the objectives of the trial. Presumably, these early deaths are rare, and would be too few to see differences between intervention and usual care practices.	cancer registry records from 2012-2018 led by Mounce (Trial Statistician) and Abel (senior statistician), of patients aged 40+ at diagnosis, 8.4% died within 30 days of diagnosis (all cause). We consider this a “significant minority”. The referee is correct that the analysis of this outcome is likely to be underpowered, but we consider it warrants inclusion.
Reviewer: 2	
2.1 Thank you for the opportunity to review this interesting work. The proposed concept sounds novel to my knowledge, and potentially promising to improve cancer diagnosis. The protocol is comprehensive, well written and largely clear. I have minor suggestions to improve clarity in parts and some thoughts regarding potential issues with the intervention and additions to the methodology that the authors may like to consider. Please note that I am unable to provide expertise regarding the statistical robustness of the protocol.	Thank you
2.2 Regarding the intervention: -If I have understood correctly, the eRAT prompt relies on data coded in the medical record. In my experience, GPs do not always code symptoms in practice management software (despite recording these in progress notes). This may lead to failure to generate a prompt, limiting the effectiveness of the intervention. I imagine the researchers are aware of this potential issue; are there measures in place to support/encourage GPs in the intervention group to consistently code patient symptoms?	This interpretation is correct, and we fully accept the point that the eRAT prompt requires good quality symptom coding. However, we deliberately have not added any ‘encouragement’ to improve coding for the intervention group. This is because if the trial showed the intervention to work, we would not be sure if it still worked without the ‘encouragement’.
2.3 For intervention practices, I wonder whether GPs may be falsely reassured if an eRAT prompt is not generated for a patient with cancer, counterintuitively leading to diagnostic delay? This study as it is planned should, however, help to answer that question.	We agree. Of course, we will not be able to identify such patients, but the overall effect will include them (if R2’s concerns are justified).
2.4 Regarding the study methodology: -Page 7, “Sample size”: I agree with the authors that the absolute reduction of 4.8% in stage 3/4 cancer diagnoses is quite large; and that it is unfortunate that detecting a smaller improvement is not feasible due to sample size required.	We agree.

2.5 Page 10, point 7 “Adverse events”: In addition to harms from investigations and psychological distress, it may be appropriate to consider the potential harms of cancer over-treatment in this analysis (see, for example: https://eur03.safelinks.protection.outlook.com/?url=https://www.racgp.org.au/download/Documents/Guidelines/prostate-cancer-screening-infosheetpdf.pdf&data=05%7C01%7CW.Hamilton%40exeter.ac.uk%7Ccf81a5fed6f0491d49ce08daef0c8ece%7C912a5d77fb984eeeaf321334d8f04a53%7C0%7C0%7C638085135105196869%7CUnknown%7CTWFpbGZsb3d8eyJWljoic4wLjAwMDAiLCJQIjoiV2luMzliLjBtIi6k1haWwLjCjXVCi6Mn0%3D%7C3000%7C%7C%7C&sdata=MMm9iWw2FEfyGMu7zxoVEUR1ZwmBTloFsY3HG1NITAE%3D&reserved=0	This is a point which we have considered in depth in our years of symptomatic cancer work. There is no question that overdiagnosis and overtreatment exist in the screening context. In the symptomatic context, it is much less likely, in that it is reasonable to assume a cancer causing symptoms is relevant to the patient’s health. Three cancer sites may still be associated with overdiagnosis even in the symptomatic arena: melanoma (melanoma numbers are increasing, yet the mortality is largely unchanged); prostate (where investigation of urinary symptoms often includes a PSA, sometimes uncovering a cancer unrelated to the symptoms); thyroid (this is largely due to indiscriminate use of ultrasound). We are studying none of these cancers. Thus, we prefer not to discuss the issue of overdiagnosis when from the above, it is likely to be very small or even absent.
2.6 In addition to comparing the stage at cancer diagnoses for all 6 eRAT cancers combined, will subgroup analysis also be performed for each of the 6 cancers individually? It would be interesting to see if there was a difference in efficacy among the 6 eRATs. For example, it may be that for some cancers symptoms occur primarily in stage 3 or 4 disease, making the intervention less effective.	We will not be powered to do site-specific analyses of the main outcome.
2.7 As early detection of malignancy does not always confer mortality benefit, the secondary outcome of survival is perhaps the most important outcome for this study. For example, a recent trial that concluded that despite earlier detection of ovarian cancer with screening (using CA125 levels or transvaginal ultrasound), this did not translate to a mortality benefit (Menon U, Gentry-Maharaj A, Burnell M, et al. Ovarian cancer population screening and mortality after long-term follow-up in the UK Collaborative Trial of Ovarian Cancer Screening (UKCTOCS): a randomised controlled trial. Lancet. 2021;397:2182-93. 33991479).	We agree with this comment, though again it is important to remember the difference between the screening and symptomatic populations (in that the prevalence of disease is generally considerably greater in the symptomatic population).
2.8 Page 22, Appendix A: There is a risk of underestimating the effectiveness of the intervention by classifying mergers of control/non-trial practices + intervention practices as intervention practices. I imagine that the authors have considered this and decided that the benefit of avoiding reduction in the study power justifies this risk?	We agree (once again) with this comment, but as surmised we took the practical decision of maximising power.
2.9 Page 23, Appendix B: Is there a rationale for the estimated 40% patient acceptance rate to respond to the survey? I wonder if this might be a little generous?	We believe 40% to be a conservative estimate, as in our experience there is a high response rates to take part in such patient questionnaires in GP practices, for example, in Banks 2014 the response rate was 71% (3469/4884) of participate in GP practices waiting room who agreed to take part in questionnaire study about cancer testing. http://linkinghub.elsevier.com/retrieve/pii/S1470204513705886

2.10 Appendix D: Will GPs be remunerated for their time to participate in training? This may increase engagement?	Thank you. We had omitted this, plus omitted remuneration for the health economic sub-study. These have been added.
2.11 Page 33, Appendix E: Does a GP consulting session in England usually include 24 back-to-back consultations without any slots deliberately remaining unbooked (for administrative work, catch up time etc)?	There is no fixed regime of appointments, with some doctors factoring in breaks in the way R2 describes, and others not doing so.
2.12 Page 34, Appendix E, Primary Outcome: The duration that a GP keeps a chart open doesn't always correlate with consultation length (some GPs keep charts open to write notes later etc).	Thank you; this is a correct observation. It forms the basis for very detailed electronic searching and analysis which we are undertaking as part of a PhD associated with this trial. There are many potential pitfalls associated with timing data and we are delineating and investigating these in detail. Our more detailed analysis plan for this sub-study includes adjustment for overly long durations of a chart being open (assuming it has been left open after a consultation has ended). Similarly adjustment will be made to very short (under 1min) durations.
2.13 Page 35, Appendix E, Measuring duration of consultations: There may be differences in complexity for patients who trigger eRATs vs those who do not that are unrelated to the eRAT prompt (ie. eRAT prompt patients may be more unwell than others and therefore require more time). Rather than comparing time of consultations with eRAT trigger vs no eRAT trigger in intervention practices, it would be more accurate to compare time of consultations where an eRAT is triggered in intervention practices with time of consultations where an eRAT would have been triggered in control practices. If this is not possible, data analysis should adjust for patient factors known to affect consultation length as well as for factors such as consulting GP, time of day etc.	The timing/workload implications of triggering an eRAT are complex. Virtually no one in the UK has tackled this problem – it is correct to note that undertaking control work is desirable, but we need first to demonstrate the feasibility of obtaining accurate timing data and, as the reviewer notes, relate this to the complexity of patient flow, and to patient complexity. As also noted, we are indeed developing models to adjust for a variety of factors such as those outlined
2.14 Points for clarification/minor edits: -Page 5, first line: I don't quite understand the 'symptom checker'. Are you saying that the computer brings up the completed eRAT checklist that generated the prompt, and the clinician checks which symptoms it has identified and manually edit this if necessary?	Exactly right! Even so, R2's uncertainty prompted us to rephrase this to 'A second function is the 'symptom checker', allowing the clinician to add additional patient's symptoms to the eRAT checklist on screen; this process automatically recalculates the risk of any of the six cancers.'
2.15 Page 5, "Justification of cancer sites": This is slightly confusing. The first line states that 18 RATs are available. The third line goes on to mention the 15 most common cancers. It isn't immediately clear (until I added up the numbers) whether the 'six cancers selected' (line 5) and 'nine remaining cancers' (line 6) are referring to the 18 RATs or the 15 most common cancers. Perhaps make a slight edit on line six to clarify – 'The remaining nine of the fifteen most common cancers were considered...'. I would also suggest listing the six cancer sites that were selected for eRATs in this section.	We have listed the selected six cancer sites, and have slightly reworded this paragraph to make it simpler to understand.
2.16 Page 8, "Practice Recruitment": The sentence beginning 'To promote practice awareness...' requires	This has been reworded. It now reads: All allocations using the minimisation algorithm

grammatical correction.	will retain a stochastic element, aimed at promoting allocation concealment (see p.7)
2.17 Page 14, “Nested Studies, Health Economics”: ‘For three cancer sites we will use decision analytic models...’ Which 3 cancer sites are you referring to?	Anne The 3 cancer sites are: colorectal cancer, lung cancer and ovarian
2.18 Page 21, “Appendix A”. Missing word in first sentence of third paragraph – “Excluding practices who restructure...”	Thank you for this: ‘practices’ added.
2.19 Page 22, “Appendix A”: ‘Where changes in list size...from one month prior to the drop’ – A change could be a drop or alternately an increase in size – please clarify whether this approach will be applied for any change (increase or decrease) or only for a drop in size?	Thank you. We have changed the word ‘drop’ to ‘change’ to reflect the fact that rises as well as falls in practice size lead to an adjustment.
2.20 Page 23, “Appendix B”: Note incomplete sentence “With a conservative estimate of a cluster size of five patients responding to the questionnaire.”	Again well spotted – a comma had been replaced by a full stop. Corrected.
2.21 Page 26, “Appendix D”: Unbalanced parentheses– ‘(i.e., ongoing use of the eRATs...’	Thank you - corrected
2.22 Page 28, “Appendix D”: Second paragraph, last line – “GPs” does not require an apostrophe -Page 29, “Appendix D”: 4th paragraph, 6th line – suggest clarification - “(false positive eRAT result)” - currently not immediately obvious whether the cancer diagnostic test or the eRAT is the false positive.	The ‘grocer’s apostrophe removed (and we found a separate missing apostrophe which we added.) For both mentions of ‘false positive’ there is an accompanying definition clarifying what the term refers to.
2.23 Page 34, “Appendix E”: Final line, remove “practices across the nested studies” (repetition) - Page 35, “Appendix E”: First line, remove “only” (repetition)	Thank you – corrected.
2.24 Thank you again for the opportunity to review this protocol. I would like to acknowledge my colleague, Dr David King, who also provided input into this review. I look forward to hearing the study outcomes.	So do we!
Reviewer: 3	
3. A well done design paper.	Thank you.
Reviewer: 4	
4.1 This review is mainly for the statistical part only. The vast majority of statistical methodology is already written well. I have some minor comments.	Thank you.
4.2 Survival analysis. The author mentioned secondary survival outcomes of 30-day, 1-year and 5-year. However, the data collection window is 1/6/2022-29/5/2024. How would the survival status be collected?	The active trial window is as the referee describes, though obtaining the routinely collected outcome data will take approximately another year and this follow-up data will be linked to from the Office for National Statistics all-cause mortality data at that point. We now describe the source in point 6. (p.9)
4.3 The definition of survival: does the author mean the overall survival or specific cancer survival?	This will be overall survival as it will be based on all-cause mortality data.
4.4 cost-effectiveness analysis (CEA). For the decision analytical model, I would like to see more details regarding the decision tree. Then, it would be helpful to understand how the QALY For probabilistic sensitivity analysis, I would like the author to provide more information: such as the distribution of costs, or what kinds of parameters would be samples.	We will use state-transition decision analytic to estimate survival and lifetime QALY gain based upon the primary outcome (full TNM staging (I-IV)) to, see appendix reference. Siebert U, Alagoz O, Bayoumi AM, Jahn B, Owens DK, Cohen DJ, et al. State-Transition Modeling:A Report of the ISPOR-

	SMDM Modeling Good Research Practices Task Force—3. Medical Decision Making. 2012;32(5):690-700 For the probabilistic sensitivity analysis we will draw on patient-level data obtained from NHS Digital, who now contain NCRAS. All the data for people who have cancer will be linkable to NCRAS and provides patient-level data including data from the Diagnostic Imaging Dataset - imaging investigations (colonoscopies, sigmoidoscopies, upper gastro-intestinal endoscopies, chest x-rays, abdominal ultrasounds, and abdominal CT scans)
4.5 Time horizon is also a key part in CEA. Can the author elaborate more on this?	The within-trial analyses will be over a 24 month period. The decision analytic model will model the impacts of any change in stage of diagnosis between the intervention and control practices over the expected lifetime of patients. The study will discount both costs and outcomes at 3.5% as recommended by the National Institute of Health and Care Excellence.

VERSION 2 – REVIEW

REVIEWER	Hayley Thomas University of Queensland Faculty of Medicine and Biomedical Sciences, Primary Care Clinical Unit
REVIEW RETURNED	23-Jan-2023

GENERAL COMMENTS	Thank you for the opportunity to review this revised manuscript, and for your thoughtful responses to the initial comments. I am satisfied with the authors' responses. I have only a couple of minor suggestions to improve clarity:  -2.6: Suggest specifying inability to do site-specific analyses of the main outcome as a study limitation. -2.12: Suggest mentioning this adjustment for length that the chart is open in Appendix E. -2.13: Suggest mentioning this as a study limitation. -2.17: Could you specify the 3 cancer sites in the article text? Thanks again, and I hope that the study goes well.
---

REVIEWER	Kuen-Cheh Yang National Taiwan University Hospital, Bei-Hu Branch, Taipei, Taiwan,
-----------------	---

	Family Medicine
REVIEW RETURNED	07-Feb-2023
GENERAL COMMENTS	All the questions were answered appropriately.